# LIFT: Efficient Layer-wIse FineTuning for Large Language Models

## Abstract

Fine-tuning is widely applied in language language processing to adapt the model for downstream tasks. However, as model sizes grow rapidly, fine-tuning the full model is computationally expensive. Conventional studies mainly focused on parameter-efficiency, but reducing the number of trainable parameters does not translate to less backward computation and fine-tuning speedup. Parameter-efficient fine-tuning still needs to perform complete backward pass to the foremost layer to calculate required gradients. For example, applying LoRA on Llama reduces the trainable parameters by $100\times$ but the fine-tuning throughput is only $1.2\times$ better. To achieve real training throughput improvement, we propose **LIFT**: a **L**ayer-w**I**se Fine-**T**uning strategy that only learns *one layer* at a time. This approach not only reduces the number of trainable parameters but also improves the finetuning throughput. We thoroughly evaluated the effectiveness of LIFT on BERT, GPT, and LLaMA models. LIFT saves the fine-tuning memory upto 3.7 $\times$ and improves the throughput by 2.1x to 2.7x compared to full fine-tuning with comparable final quality. We further shows that LIFT is orthogonal with existing methods like LoRA and combining them can achieve both compute-efficiency and parameter-efficiency.

## 1 Introduction

Large pre-trained transformer-based language models, particularly bidirectional masked language models from the BERT family (Devlin et al., 2019; Liu et al., 2019; 2021a), have led to significant improvements in many natural language processing tasks. These models are pre-trained on large, annotated corpora using the language modeling objective, and then fine-tuned on task-specific supervised data, resulting in impressive achievements in various domains (Radford et al., 2022; Copilot; Chowdhery et al., 2022). However, the large size of these models makes them prohibitively expensive to train and fine-tune, preventing researchers to explore efficient methods for adapting the models to different scenarios.

Existing efficient fine-tuning methods mostly focus on reducing the number of *learnable parameters*, but pay less attention to the measured *fine-tuning throughput*. Existing studies attempt either to adapt only a small number of parameters (Zaken et al., 2021b), or learnable soft prompts (Liu et al., 2021a; Li & Liang, 2021b), or tiny external modules (Hu et al., 2021; Houlsby et al., 2019; Liu et al., 2022) for new tasks. While the number of learned and stored parameters can be significantly reduced, the back-propagation cost keeps high and un-optimized.

**Parameter-efficiency does NOT necessarily translate to measured speedup**, and the measured speedup is what the community should focus on. For example, when using Adapter (Houlsby et al., 2019), which has 275 $\times$ fewer trainable parameters, the fine-tuning throughput is only 1.25 $\times$ better than fine-tuning the full model. To provide a more concrete example, consider fine-tuning BERT with 12 blocks. Despite the 1st and 12th blocks having the same number of parameters, the cost to calculate corresponding gradients is drastically different due to the location and depth of learnable parameters. The 1st block requires back-propagation all the way back to *the very first layer*, while the 12th block only needs to back-propagate to *one layer before*. Thus same number of learnable parameters can result in very different back-propagation costs, which reveals that simply learning less parameters does not always lead to higher training throughput.

**Table 1.** The comparison between existing fine-tuning methods focus on parameter efficiency.

| Method | Storage | Peak Memory Saving | | Backprop FLOPs Saving | No Inference Overhead |
|---|---|---|---|---|---|
| | | Optim State | Attn | | |
| Adapter (Houlsby et al., 2019) | yes | yes | no | no | no |
| Prefix (Li & Liang, 2021a) | yes | yes | no | no | no |
| BitFit (Zaken et al., 2021b) | yes | yes | yes | no | no |
| LST (Sung et al., 2022) | yes | yes | yes | yes | no |
| AutoFreeze (Liu et al., 2021b) | no | no | no | yes | yes |
| LoRA (Hu et al., 2021) | yes | yes | yes | no | yes |
| LIFT(ours) | yes | yes | yes | yes | yes |

To mitigate the gap and bring fine-tuning throughput improvements, we propose **L**ayer-w**I**se **F**ine **T**uning (**LIFT**) approach: which **only updates one layer** in each iteration and freezes the rest of the parameters. LIFT allows us to train language models only to a shallower location (x mark in Figure 1.*Right*) while existing methods require back-propagation to the foremost layer, even when the number of learnable parameters is much smaller Figure 1.*Left*). Such a learning scheme not only reduces the number of learnable parameters (since most layers are frozen), but also leads to less backward computation (by reducing the back-propagation depth to half), resulting in a measured speedup. Unlike previous studies where most parameters are forever frozen during the fine-tuning process, LIFT optimizes parameters in an iterative manner, allowing every parameter to be updated We demonstrate that learning one layer at each iteration does not result in inferior quality, while reducing memory and computation costs.

In this paper, we show that LIFT has several key advantages:

- LIFT neither modifies the pre-trained architecture nor increases the prompt sequence length. LIFT can be generally applied to various language models without adding inference latency.
- LIFT reduces the training memory while keeping each layer has a chance to be updated. This allows LIFT to maintain competitive accuracy with full fine-tuning.
- LIFT not only saves up to 3.7 times memory, but also increases throughput by 2.0 to 2.6 times, making fine-tuning more efficient and lowers barriers to fine-tune large models.
- LIFT is orthogonal to many prior methods and can be combined with many of them (e.g., prefix-tuning (Li & Liang, 2021a), LoRA (Hu et al., 2021)) to further boost performance and reduce storage.

We show that LIFT is surprisingly effective on various models and tasks, and has a large practical utility in fine-tuning models in memory- and compute- constrained environments. We also perform ablation studies of different LIFT settings to provide insights for future efficient algorithm design.

## 2 RELATED WORK

### 2.1 TRANSFORMER LANGUAGE MODELS

The Transformer architecture (Vaswani et al., 2017) heavily relies on self-attention mechanisms in its sequence-to-sequence design. When it was first proposed, (Radford et al., a) applied this model to language modeling by stacking of Transformer decoders, consisting of multi-head attention (MHA) followed by fully-connected networks (FFNs).Since then, Transformer-based language models have become the dominant approach in natural language processing, achieving state-of-the-art results in various tasks. A new methodology emerged with the release of BERT (Devlin et al., 2019) and GPT-2 (Radford et al., b) - both of which are large Transformer models pre-trained on vast amounts of text - that demonstrates a significant performance boost when fine-tuned on task-specific data, as opposed to training on such data from scratch. Increasing the size of Transformer models tends to result in improved performance and is an ongoing area of research. Till now, GPT-3 (Brown et al., 2020) and OPT (Zhang et al., 2022) hold the largest single transformer language model size of $\sim$ 170B parameters[*]. The tremendous model sizes and extensive resource requirements raise the need to tune transformers *efficiently* to adapt to different downstream tasks.

---

[*]GPT-4 technical report refrained from specifying the model size.

## 2.2 PARAMETER-EFFICIENT FINE-TUNING (PEFT)

Parameter-Efficient Fine-Tuning (PEFT) methods aim to enable efficient adaptation without fine-tuning all the model's parameters. PEFT methods can be classified in multiple ways.

**Selective Methods** propose fine-tuning only a subset of models. For instance, (Zaken et al., 2021a; Cai et al., 2020) analyzed the bias terms and concluded that fine-tuning only the bias terms can be as competitive as fine-tuning the entire model. However, this method is no longer competitive and shows inferior performance when the dataset size grows (Zaken et al., 2021a). Instead of learning a static set of parameters, researchers have also experimented with learning dynamic parts. FreezeOut (Brock et al., 2017) proposes gradually freezing layers and excluding front layers from the backward pass to accelerate training. Later, AutoFreeze (Liu et al., 2021b) verified the effectiveness of this approach on language models. However, these methods still require a significant amount of computation during the starting stage and result in inferior final performance.

**Additive Methods** add new layers to models instead of updating existing parameters, and only these additional layers are learned during fine-tuning (Houlsby et al., 2019; Rebuffi et al., 2017; Lin et al., 2020; Hu et al., 2021). Existing approaches, such as adapters (Houlsby et al., 2019), added the layers sequentially, resulting in increased latency during inference. LoRA (Hu et al., 2021) attempted to address this issue by merging the learned weights into the main model. Later, IA$^3$ (Liu et al., 2022) introduced novel ways to add parameters with parameter-accuracy trade-offs and LST (Sung et al., 2022) proposes a highway structure and only learn the tiny side channels to reduce memory usage. Additive methods requires manual design and many of them do not save backward computation FLOPs (IA$^3$, LoRA, Adapter). Further, sometime they even bring forward overhead (e.g., Adapter, LST) thus less preferred in practice.

**Prompt-Based Methods** suggest optimizing the input word embeddings instead of fine-tuning and aim to control the behavior of a language model by modifying the input text. They design continuous and differentiable forms of prompt engineering (soft prompt) to ease optimization. Soft prompts can be trained for the input layer only (Liu et al., 2021a; Lester et al., 2021) or for all layers (Li & Liang, 2021b). However, these approaches either use up available model input length or introduce extra inference overheads. Furthermore, they cannot reduce back-propagation costs as well.

## 2.3 GREEDY LAYER-WISE UNSUPERVISED LEARNING

Another related method is greedy layer-wise unsupervised learning (Bengio et al., 2006; Hinton et al., 2006), which trains one layer at a time for Deep Belief Networks (DBN) in the pre-deep-learning era. It is shown to be effective as an initialization for deep supervised architectures, which accelerates the training when there were no General-Purpose GPU accelerators. We recognize the potential of this idea and realize such Block Coordinate Descent based optimization can be applied to supervisely fine-tune large language models. We emphasize that the supervised layer-wise learning that we propose is distinct from early unsupervised layer-wise learning: instead of introducing extra modules as in previous work, we maintain the complete forward pass but simplify the backpropagation to only update one layer. Through our experiments, we not only show that layer-wise learning can efficiently fine-tune large language models with measured fine-tuning throughput, but we also find that different learning patterns can significantly affect the performance (in Appendix), calling for further analysis and attention on layer-wise learning.

## 3 METHOD

### 3.1 PRELIMINARIES

Consider a pre-trained language model $\mathcal{F}_\Theta\left(\cdot\right)$ with a set of parameters $\Theta = \{\mathbf{W}_1, \mathbf{W}_2, \ldots, \mathbf{W}_n\}$. Here, $W_i$ refers to the parameters in the $i$-th layer among a total of $n$ layers. $\mathcal{F}_\Theta$ can be a multi-task learner like GPT (Radford et al., b; Brown et al., 2020), which is based on the Transformer architecture (Vaswani et al., 2017). This pre-trained model can be adapted to downstream tasks such as summarization, question-answering, machine reading comprehension (MRC), and description to sentences (E2E). Each downstream task is represented by a training dataset of context-target pairs, denoted by $\mathcal{Z} = \{(x_i, y_i)\}_{i=1,..,N}$, where both $x_i$ and $y_i$ are sequences of tokens. For example, in

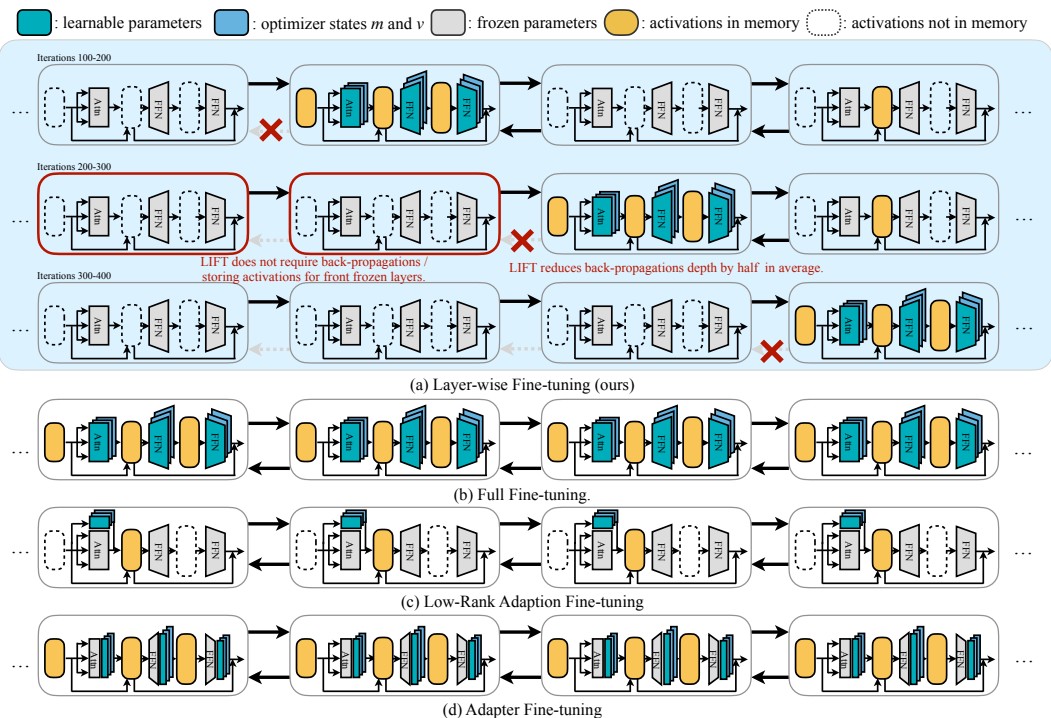

**Figure 1.** The overview of LIFT, and its comparison between LoRA (Hu et al., 2021), Adapter (Houlsby et al., 2019) and Fine-tune Full network (FT-Full for short). In LIFT, there is only one layer (Transformer Block) being updated in each iterations, therefore, the back-propagation can be stopped in the middle thus reduces the back-propagation depth by half (in average). Existing methods focus mainly on parameter-efficiency but still have to back-propagate to the very first layer. This leads to high computation costs as the back-propagation FLOPs is $2 \times$ larger compared to the forward.

E2E, $x_i$ represents the information and $y_i$ represents its corresponding restaurant description. In summarization, $x_i$ is the content of an article and $y_i$ is its summary.

**Vanilla Full Fine-Tuning.** During full fine-tuning, the model is initialized with pre-trained weights $\Theta_0$ and updated to $\Theta_0 + \Delta\Theta$ by repeatedly following the gradient to optimize the conditional language modeling objective.

$$\min_{\Theta} \sum_{(x,y)\in\mathcal{Z}} \sum_{t=1}^{|y|} \mathbb{L}\left(y_t, F_{\Theta}(x, y_{<t})\right) \tag{1}$$

W $\Delta\Theta = \nabla\mathcal{F}_{\Theta}(\mathbf{X})$, where $\mathbf{X} \in \mathcal{Z}$ and $\mathcal{Z}$ is the training dataset. Since all parameters are updated, the model has the greatest number of learnable parameters, $\|\Delta\Theta\| = \|\Theta\|$ thus delivers the best model quality. However, this also results in the largest training cost: the back-propagation is performed all the way to the foremost layer and Adam-based (Kingma & Ba, 2014) optimizer needs to keep first- and second- momentum buffers to stabilize convergence ($2 \times$ as large as the model size $\|\Theta\|$).

**Parameter-Efficient Fine-Tuning.** For common approaches like LoRA (Hu et al., 2021) and Adapter (Houlsby et al., 2019), they propose to learn a much smaller number of parameters

$$\min_{\Theta} \sum_{(x,y)\in\mathcal{Z}} \sum_{t=1}^{|y|} \mathbb{L}\left(y_t, F_{\Theta_0+\Phi(\Theta)}(x, y_{<t})\right) \tag{2}$$

When $\|\Phi(\Theta)\| << \|\Theta\|$, learning fewer parameters reduces the memory required for optimizer states. However, the intermediate activations may still be high, as shown in Figure 1, and become more expensive as batch size increases. Existing PEFT methods distribute learnable parameters evenly through the model, as in $\{\mathbf{\Phi}(W_1), \mathbf{\Phi}(W_2), \dots, \mathbf{\Phi}(W_n)\}$, therefore back-propagation still needs to go through the foremost layer, and required computation is not improved.

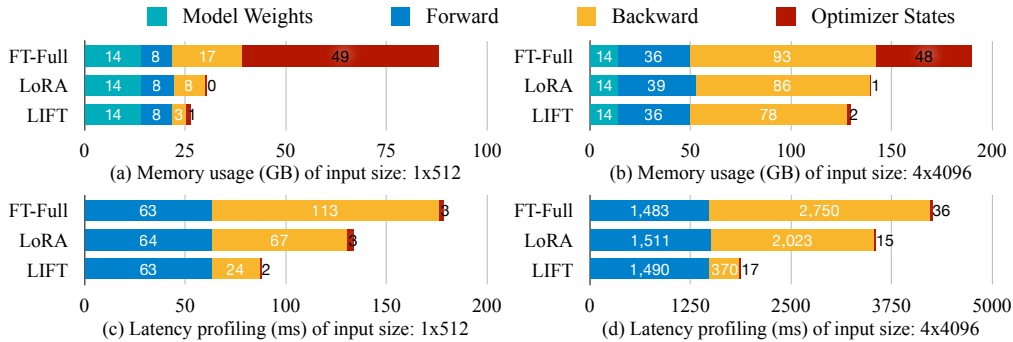

**Figure 2.** The memory and latency profiling results for Llama-7B with different batch-size and sequence length. LIFT improves both backward memory and latency thus significantly improving fine-tuning throughput, while conventional PEFT methods only reduce the number of learnable parameters.

## 3.2 Layer-Wise Fine-Tuning (LIFT)

In order to make it easier for more devices to use expensive language models, we propose a new fine-tuning system that reduces back-propagation costs without sacrificing parameter-efficiency. In this section, we will introduce our LIFT algorithm, which is both simple and effective. It offers memory saving, latency reduction, and high usability and versatility. We will discuss the key elements and provide a step-by-step methodology for designing and implementing LIFT.

The core idea of our proposed LIFT is "**learning one layer/block at a time**" (shown in Figure 1, *Right*). For a given pre-trained language model $\mathcal{F}_\Theta(\cdot)$ with a total of $N$ blocks, let the set of all transformer blocks be $\mathcal{B} = \{\mathbf{B}_1, \ldots, \mathbf{B}_N\}$. In each iteration, LIFT first performs the normal forward pass. During the backward pass, LIFT selects one layer $\mathbf{B}_\ell \in \mathcal{C}$ as the candidate, and only the selected candidate is updated/learned.

During the fine-tuning process of LIFT, a single candidate is selected and updated. The selection policy can be either (i) front to end, (ii) end to front, or (iii) random. For simplicity and generality, we set the order to be *front-to-end*, as shown in the right side of Figure 1. In Appendix, we provide more detailed ablation and compare different layerwise fine-tuning schemes. LIFT optimizes each block for the *same* number of iterations within an epoch. For the first batch of iterations, the 1st block is fine-tuned, and back-propagation requires to be performed until the foremost layer. Then, LIFT selects the 2nd block, and the back-propagation depth is reduced by one, and similarly for later steps.

Unlike previous PEFT methods, which have a fixed set of learnable parameters and most of the parameters are forever frozen, LIFT allows each parameter to have a chance to be updated and learned during fine-tuning. This enhances the learning capacity and leads to better performance (see Section 4). Proper implementations ensure that this does not result in extra memory overhead. When switching from one selected block to another, LIFT can offload the first- and second-moment terms for the untrained blocks to the CPU and recover them when needed. The offloading and loading can be pipelined with training, and the overhead can be covered. Thus, LIFT can ensure convergence without keeping all moment terms in memory.

### 3.3 Analysis of LIFT's Saving

We next analyze and discuss why and how LIFT are more memory- and computation- efficient.

**PEFT Only Recues the Optimizer States**    Consider the widely used Adam-based optimizers
$$g_t = \nabla F(\theta_t); \quad v_t = \beta_1 v_{t-1} + (1 - \beta_1) g_t; \quad m_t = \beta_1 n_{t-1} + (1 - \beta_1) g_t^2 \tag{3}$$
When $|v|$ and $|m|$ have the same size of learnable parameters, vanilla full-finetuning requires storing all first- and second momentum, resulting in a $2\times$ increase in memory cost. This becomes a bottleneck when fine-tuning resource-constrained devices, especially given the large parameter size of language models. Figure. 2 (a) shows that optimizer states and model weights alone dominate more than 70% of the memory. Applying LoRA can significantly improve memory usage.

However, simply reducing the number of learnable parameters does not result in a practical speedup. As shown in Figure. 2.(C), even if LoRA only optimizes 1% of the parameters, the backward process (yellow parts) remains computationally expensive. This is because regardless of whether LoRA is used or not, back-propagation still needs to be performed on the very first layer, as the learnable parameters are evenly distributed across the model. As the batch size increases and the sequence length grows, the benefits of existing PEFT methods in terms of memory (Figure. 2.(B)) and latency (Figure. 2.(C)) become less significant.

**LIFT Cuts Back-propagation Depth by Half and Speedup Fine-tuning**    Different from conventional methods that require back-propagation through all layers, LIFT only needs to back-propagate the SELECTED block. The backward pass can safely stop since there are no learnable parameters in the front blocks.

This significantly reduces the amount of backward computation and improves the fine-tuning through-put, as backward computation is usually twice as large as forward computation. By applying LIFT, the back-propagation depth is halved on average, reducing computational cost by 1/2 during the backward pass. Additionally, computation for gradients of weights for FFNs after selected layers is skipped, further reducing backward FLOPs by 1/3. Overall, LIFT reduces backward computation by 2/3 and results in significant savings (Figure. 2 (c, d)). These improvements are not limited by the batch size or sequence length (Figure. 2 (b, d)).

**LIFT is Parameter-Efficient and Freezing Layers Saves Memory**    The first memory saving comes from optimizer states. Only the one selected block in LIFT is learnable, and other optimizer-related buffers do not need to be in memory. Therefore, LIFT also benefits from fewer learnable parameters and saves memory.

The second memory saving comes from intermediate activations. Take the form of a linear layer:

$$\mathbf{Y}^\ell = \mathbf{W}^\ell \mathbf{x} + \mathbf{b}^\ell \tag{4}$$

where the corresponding gradients are

$$\frac{\partial L}{\partial \mathbf{x}} = \frac{\partial L}{\partial \mathbf{Y}^\ell} \mathbf{W}^\ell; \quad \frac{\partial L}{\partial \mathbf{W}^\ell} = \mathbf{x}^T \frac{\partial L}{\partial \mathbf{Y}^\ell}; \quad \frac{\partial L}{\partial \mathbf{b}^\ell} = \frac{\partial L}{\partial \mathbf{Y}^\ell} \tag{5}$$

For learnable blocks, all layers within a single block $\mathbf{B}_\ell$ and all activations $\mathbf{x}$ must be saved in memory for fine-tuning. This results in crucial memory overhead especially when the batch size grows. For frozen layers, the activations $\mathbf{x}$ are no longer required, as we only need $\frac{\partial L}{\partial \mathbf{x}}$ to keep the chain rule for back-propagation. For attention layers, the backward $\frac{\partial L}{\partial x}$ is not activation-free, thus *all* attention outputs need to be stored in order to perform backward in conventional PEFT methods (Figure 1.(b, c,d)). With LIFT, only blocks after selected layers need to store the attention outputs, as front layers do not require gradients (Figure 1.(a)). We will expand and discuss the real-measured speedup and savings in Section 4.5.

## 4 EXPERIMENTS

To evaluate the effectiveness , we thoughtfully benchmark LIFT on various sized models, including BERT (Devlin et al., 2018), OPT (Zhang et al., 2022) and LLaMA (Touvron et al., 2023). Our experiments cover a wide range of natural language understanding (NLU) and generation (NLG) tasks, including the GLUE (Wang et al., 2018), QA benchmarks, and Stanford Alapaca (Taori et al., 2023). We first compare the accuracy of LIFT and other methods then benchmark the accuracy. All experiments were conducted using NVIDIA A100 GPUs, PyTorch 2.0. The Transformers version (Wolf et al., 2020) is 4.30 and PEFT verison is 0.4.0. We will release our codebase for when less anonymous.

### 4.1 BASELINES

To compare with other baselines, we follow the setups from prior work and reuse their reported numbers when possible. We focus on the comparison with LoRA and its variants as these methods introduce no inference overhead.

**Fine-Tune Full (FT-Full)** is a common approach for adaptation. During fine-tuning, the model is initialized with pre-trained weights and biases and all model parameters perform gradient updates. In our experiments, we report the number of learning all layers.

**Table 2.** Performance comparison of BERT (Devlin et al., 2018) on GLUE benchmark (Wang et al., 2018). Each result is averaged from 3 runs. For each task, the numbers highlighted in red and blue indicate the best and second-best respectively. LIFT shows on-par performance with FT-Full while existing methods sometimes suffers from performance degration.

| Method | Avg. | GLUE Benchmark | | | | | | |
|---|---|---|---|---|---|---|---|---|
| | | CoLA | MNLI | MRPC | QNLI | QQP | RTE | SST-2 |
| FT-Full | 80.2 | $58.3_{\pm1.1}$ | $83.7_{\pm0.1}$ | $82.6_{\pm0.7}$ | $90.7_{\pm0.1}$ | $90.7_{\pm0.1}$ | $63.4_{\pm2.1}$ | $91.6_{\pm0.7}$ |
| Adapter | 78.4 | $51.8_{\pm4.1}$ | $79.2_{\pm0.4}$ | $84.3_{\pm1.3}$ | $88.6_{\pm0.2}$ | $85.3_{\pm0.2}$ | $68.2_{\pm1.7}$ | $91.0_{\pm1.1}$ |
| BitFit | 78.1 | $51.1_{\pm0.5}$ | $78.6_{\pm0.8}$ | $83.6_{\pm2.6}$ | $88.5_{\pm1.0}$ | $86.0_{\pm0.1}$ | $67.9_{\pm3.3}$ | $90.7_{\pm1.3}$ |
| LoRA | 79.0 | $56.5_{\pm0.7}$ | $84.1_{\pm0.3}$ | $81.4_{\pm1.0}$ | $90.2_{\pm0.3}$ | $84.1_{\pm0.1}$ | $65.0_{\pm2.2}$ | $91.3_{\pm0.5}$ |
| LIFT | 79.5 | $56.9_{\pm1.9}$ | $83.2_{\pm0.4}$ | $81.9_{\pm1.2}$ | $89.6_{\pm0.4}$ | $88.3_{\pm0.1}$ | $63.5_{\pm2.3}$ | $91.4_{\pm0.7}$ |

**Adapter Tuning** is originally proposed as a parameter-efficient approach that inserts extra layers between the original modules and only learn the tiny extra modules to adapt to downstream tasks (Houlsby et al., 2019). We follow the original design and append layers after both attention and MLP modules in our experiments for BERT.

**Prefix Tuning** draws its inspiration from prompting and only updates virtual tokens prepended to the input (Li & Liang, 2021a). It has been proved effective for LLMs on a variety of NLG tasks. In our experiments, we follow the original prefix tuning setting for OPT and LLaMA on QA benchmarks, and the LLaMA-Adapter implementation (Zhang et al., 2023) for instruction-tuning.

**Bias-only Tuning** is a simple variant of fine-tuning where only the bias vectors are trained while everything else is frozen. We referenced the implementation from BitFit (Zaken et al., 2021a).

**Low-Rank Adaptions (LoRA)** is a method that factorizes the large weight matrix into lower ranks and only learns the low-rank parts during fine-tuning (Hu et al., 2021). We use the official settingand Huggingface's implementation to run our experiments.

**Layer-Wise fine-tuning (LIFT, ours)** does not introduce any additional modules to the original architecture and only alters the way of updating weights during fine-tuning. In the followings, we will first show that our LIFT can achieve performance that is comparable to previous fine-tuning methods. Note that LIFT is orthogonal with most PFET methods, We also evaluate the performance by combing LIFT and LoRA to show the generalization ability of LIFT. We then demonstrate the efficiency improvement on various model sizes and workloads.

## 4.2 PERFORMANCE ON NATURAL LANGUAGE UNDERSTANDING (GLUE)

For natural language understanding (NLU) tasks, we use the pre-trained BERT-base (110M) models from the HuggingFace Transformers library (Wolf et al., 2020) and evaluate the performance of different efficient adaptation approaches on tasks from the GLUE benchmark. we set learning rate at 1e-4, weight decay to be 0, sequence length at 256 and batch size at 32. To make a fair comparison, we train all settings **same epochs** at 3 and no extra epochs are allocated to LIFT. All 12 blocks in BERT are fine-tuned from front to end in each epoch.

Based on Table 2, we observe that while conventional efficient learning approaches sacrifice performance to some degree, LIFT closely matches the baseline performance (FT-Full) and consistently outperforms other efficient fine-tuning methods on several benchmarks. Notably, on the QQP task, LIFT improves performance by 3~4% in comparison with existing PEFT approachs.

## 4.3 PERFORMANCE ON QA BENCHMARKS

We next evaluate the LIFT's performance on larger OPT (Zhang et al., 2022) and LLaMA (Touvron et al., 2023) models. We include zero-shot as a new baseline and replace the adapter setting (Houlsby et al., 2019) with prefix tuning (Li & Liang, 2021a) as the latter one's performance is more competitive on large language models. We reference the setting from (Wu et al., 2023) and set to learning to 5e-6 for FT-Full and 5e-5 for other settings. We use the standard warmup strategy and cosine annealing decay to adjust learning during 3 training epochs.

**Table 3.** Performance comparison of OPT-1.3B (Zhang et al., 2022) and LLaMA-7B (Touvron et al., 2023) on QA benchmarks. The best and second-best results are highlighted with color. While LIFT demonstrates competitive performance, the performance improvement is significantly larger on challenging tasks (where zero-shot cannot handle) compared other PEFT methods, as no parameters are forever frozen in LIFT .

| Language Model | Method | QA Benchmarks | | | | | | |
|---|---|---|---|---|---|---|---|---|
| | | PIQA | HellaSwag | SciQ | OpenBookQA | WebQs | ARC-e | ARC-c |
| OPT-1.3B | Zero-shot | 72.5 | 41.5 | 84.4 | 23.4 | 4.7 | 57.0 | 23.4 |
| | FT-Full | $75.6_{\pm1.2}$ | $47.3_{\pm2.7}$ | $91.7_{\pm0.3}$ | $37.2_{\pm3.1}$ | $34.8_{\pm1.1}$ | $61.7_{\pm0.7}$ | $31.4_{\pm2.2}$ |
| | LoRA | $73.5_{\pm1.1}$ | $42.8_{\pm3.3}$ | $93.7_{\pm0.2}$ | $26.4_{\pm4.4}$ | $19.8_{\pm1.9}$ | $59.7_{\pm0.7}$ | $28.1_{\pm1.7}$ |
| | LIFT | $73.9_{\pm1.4}$ | $43.1_{\pm2.9}$ | $92.6_{\pm0.2}$ | $29.0_{\pm5.7}$ | $27.5_{\pm1.9}$ | $60.1_{\pm0.9}$ | $27.8_{\pm2.2}$ |
| | LIFT + LoRA | $73.8_{\pm1.7}$ | $44.7_{\pm3.2}$ | $91.8_{\pm0.7}$ | $27.4_{\pm4.0}$ | $17.8_{\pm3.1}$ | $60.1_{\pm2.2}$ | $27.1_{\pm2.9}$ |
| LLaMA-7B | Zero-shot | 77.4 | 56.4 | 89.7 | 28.2 | 0.0 | 67.3 | 38.2 |
| | FT-Full | $82.4_{\pm0.8}$ | $59.4_{\pm2.1}$ | $95.6_{\pm0.3}$ | $47.8_{\pm5.5}$ | $44.2_{\pm1.7}$ | $77.4_{\pm0.4}$ | $49.9_{\pm1.5}$ |
| | LoRA | $81.6_{\pm0.9}$ | $59.8_{\pm2.5}$ | $96.2_{\pm0.5}$ | $38.0_{\pm4.7}$ | $33.1_{\pm2.0}$ | $74.5_{\pm0.3}$ | $40.2_{\pm2.0}$ |
| | LIFT | $81.1_{\pm1.1}$ | $60.4_{\pm1.9}$ | $96.3_{\pm0.4}$ | $37.6_{\pm4.9}$ | $44.1_{\pm2.1}$ | $75.8_{\pm0.3}$ | $46.8_{\pm2.1}$ |
| | LIFT + LoRA | $81.4_{\pm1.0}$ | $61.3_{\pm2.23}$ | $95.9_{\pm0.5}$ | $37.1_{\pm4.0}$ | $39.7_{\pm2.5}$ | $74.7_{\pm0.7}$ | $39.9_{\pm1.8}$ |

**Table 4.** Performance comparison of LLaMA (Touvron et al., 2023) on Stanford Alpaca instruction-tuning dataset (Taori et al., 2023). The GPT-4 score is evaluated following Vicuna's (Chiang et al., 2023) pipeline where the comparison reference is *chatgpt-3.5-turbo*.

| Language Model | Method | Latency (ms) | Memory (GB) | Loss ($\downarrow$) | GPT-4 Score ($\uparrow$) | Reference Score |
|---|---|---|---|---|---|---|
| LLaMA-7B | FT-Full | 514 | 91 | 0.34 | 474 | 608 |
| | Adapter | 476 | 42 | 0.44 | 398 | 628 |
| | LoRA | 403 | 41 | 0.41 | 404 | 616 |
| | LIFT | **221** | **32** | 0.36 | 470 | 614 |
| LLaMA-13B | FT-Full | 892 | 166 | 0.25 | 505 | 594 |
| | Adapter | 704 | 71 | 0.38 | 474 | 624 |
| | LoRA | 674 | 68 | 0.31 | 486 | 610 |
| | LIFT | **365** | **57** | 0.28 | 494 | 608 |

Evaluation on OPT-1.3B and LLaMA-7B are conducted on seven QA benchmarks and the results are attached in Table 8. While LIFT can in many cases outperform previous approaches, we notice that the performance gap is more obvious on challenging tasks as WebQs, ARC-e, and ARC-c, where zero-shot fails to yield high quality answers. This suggests that LIFT has higher fine-tuning capacity as all parameters have a chance to be updated, while the most parameters are forever frozen in conventional methods. We also attach the corresponding validation curve in Appendix.

Furthermore, since LIFT is a simple and general method, we present the results of combining LIFT and LoRA in the last rows of Table 8. This combination yields answers of comparable quality and suggests that we can leverage both the speedup in fine-tuning from LIFT and the storage savings from LoRA.

## 4.4 PERFORMANCE ON INSTRUCTION TUNING

With growing attention ChatGPT (OpenAI, 2022) has received, the demand for to tailoring the model to their domain-specific needs (e.g., Law, Biomedical, Health Care) has been increasing rapidly. We start with LLaMA-7B and LLaMA-13B (Touvron et al., 2023) and align pretrained language models with instructions following the self-instruct (Wang et al., 2022) and using data from Stanford Alpaca (Taori et al., 2023). We train the models with batch 2, accumulation steps 8 and learning rate 2e-5 for FT-Full and 4e-4 for other methods. All results are obtained from training with 3 epochs.

For evaluation, we report the fine-tuning and loss and GPT4-score, following Vicuna (Chiang et al., 2023) setting to use GPT-4 as the automated evaluation using 80 provided questions. The quality of the answers are evaluated based on helpfulness, relevance, accuracy, and details. This is an pair-to-pair comparison and we choose *ChatGPT-3.5-turbo* as the baseline in our experiments.

Table 4 shows that LIFT generates close-quality answers as FT-Full, while other methods like LoRA and Adapter more or less suffer from performance degradation in term of training loss when aligning

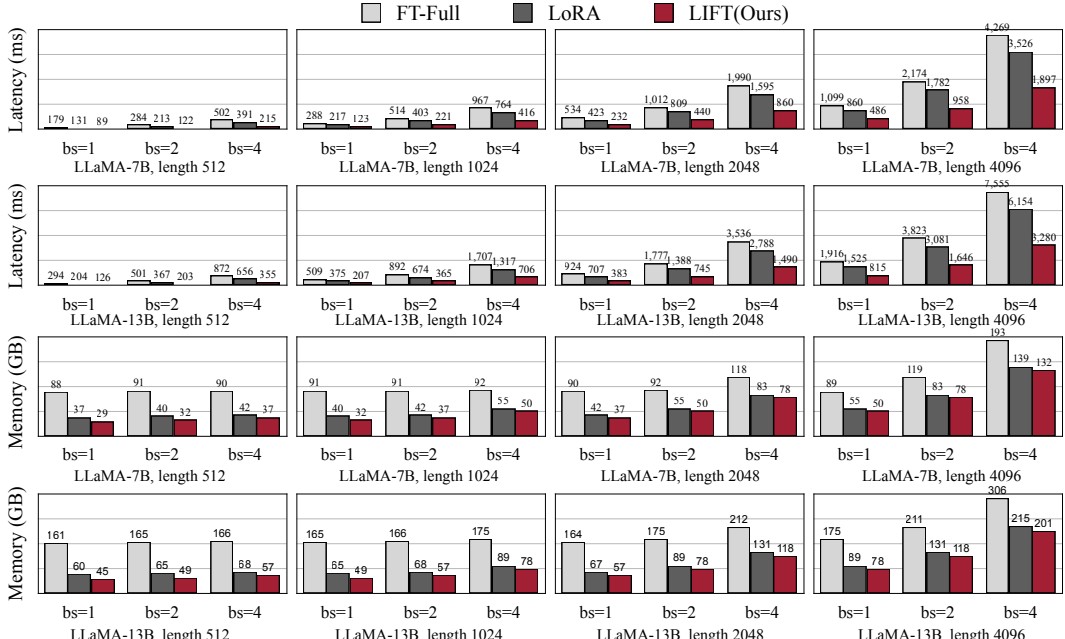

**Figure 3.** The compute-, and memory-efficiency comparison for LIFT on different sized models with varying length settings. We report the average latency and peak memory. LIFT significantly improves training throughput while using lower memory footprint than previous methods. The improvement is more significant for large models and batch sizes.

with instructions. For GPT-4 score, the trends is also similar: LIFT has better answer quality and the gap between LLaMA generated answers and ChatGPT reference are smaller.

### 4.5 EFFICIENCY COMPARISON

We have demonstrated that LIFT can be a viable alternative to prior methods, and now further investigate its efficiency in comparison to other fine-tuning techniques. We will be examining two key aspects - the training throughput and the peak memory footprint. These numbers were measured on Eight Nvidia A100 (80G) SXM GPUs using the Accelerate's library auto parallelism.

We thoughtfully profile LIFT on various models sizes and compare the results in Figure. 3. Conventional wisdom suggests that fewer learnable parameters would always result in lower training memory and computation. But in first two rows of Figure. 3, we find that the latency saving from existing methods is limited. From the last two rows of Figure. 3, we notice that even the memory reduction would saturate the fine-tuning memory is no longer dominated model weights and optimizer states.

In contrast, LIFT only propagates gradients until the selected layer, avoiding unnecessary backward computations on front layers and reducing the computation required by calculating weight gradients in later FFN layers, thus achieving consistent and measured speedup compared with previous methods. The saving ratio does not saturate when fine-tuning workloads increases (LLaMA-7B and 13B latency results), while peak memory usage keeps similar with existing PEFT method.

### 5 CONCLUSION

We propose LIFT, a cost-effective method for fine-tuning language models that can reduce back-propagation computation. LIFT does not neither change the model architecture nor reduces input sequence length, and preserves high model quality while improves the training throughput up to 1.8 to 2.7× by reducing back-propagation costs. The saving becomes more significant with larger model sizes and batch sizes. LIFT accelerates fine-tuning process and thereby democratize language models to a wider range of scenarios.

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
