## A  BROADER IMPACT

In this paper, we propose a technique that can make it easier to fine-tune large language models (LLMs) in an efficient manner. This approach enables more people to benefit from LLMs, both in positive and negative ways. On the positive side, our work aims to make LLMs more widely accessible, which can help to democratize access to these models and encourage more people to generate new ideas, especially for those with limited resources / budgets. It also reduces the costs associated with fine-tuning LLMs and enables more fine-tuning to be performed locally, which may help to address concerns around data privacy (since data no longer needs to be sent to the cloud). However, customized LLMs can be exploited by malicious users to spread misinformation, commit fraud, and reinforce biases if not properly monitored. Our proposed method also makes fine-tuning LLMs more accessible on the dark side.

## B  RANGE SELECTION AND CONTRIBUTION ANALYSIS

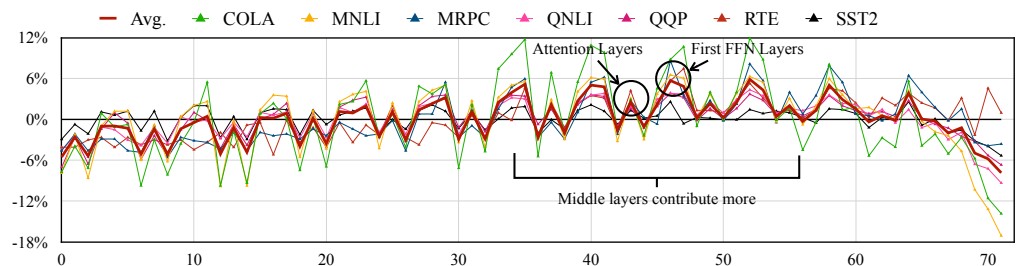

**Figure 4.** Layer-wise contribution analysis on BERT (Devlin et al., 2018) for different downstream tasks on GLUE (Wang et al., 2018). Not all layers contributed equally to the accuracy. Attention and first-FFN hae a higher impact, and middle layers in general contribute more to the performance.

While LIFT has proven that fine-tuning one block at a time can achieve ideal performance, it is natural to wonder if all blocks contribute equally to the downstream task. To investigate this, we select one block for each plot and fine-tune only the chosen block and the last classifier until convergence. By enumerating combinations of all layers (from embedding to classifier) and datasets, we plot Figure. 4 and make several interesting observations.

- The performance of all downstream tasks on GLUE are *highly correlated*.
- *Attention and first-FFN* contribute more to the downstream performance.
- *Middle blocks* of BERT are more important in evaluations of all GLUE tasks.

These findings suggest that LIFT may reduce the number of blocks and focus on optimizing important blocks to further boost performance. We evaluated the performance of LIFT with different block selection ranges and present the results in Table 5. The contribution score for each layer (including Q, K, V, FFN linear layers, and layer-norm layers) corresponds to the accuracy of fine-tuning that layer and the classifier head. All results are relative numbers with respect to the average accuracy across all layers. We aggressively scan the learning rate to minimize the effects of hyper-parameters. The results indicate that the best-performing range is the middle 3 blocks $[6, 8]$. Training only the first or last 3 blocks, such as $[1, 3]$ and $[10, 12]$, results in significantly worse performance. These findings are consistent with the contribution analysis shown in Figure 4.

However, this trend no longer holds when experiments are scaled up and models start to demonstrate emergent abilities. As shown in Table 6, updating all layers with LIFT consistently demonstrates better performance than other range selections. Therefore, we did *not* restrict the range of blocks to keep LIFT simple and general. In our main experiments, we set LIFT to iterate *all layers*.

## C  TRAINING SCHEDULE OF LIFT

In addition to layer range, it is also worth discussing the schedules with which the LIFT is trained. The schedules can be *end2front*, *front2end*, or *random*. For the first two schedules, all candidates are

**Table 5.** Ablation studies on layer range selection with BERT (12 blocks) (Devlin et al., 2018). The middle blocks appear to play a more critical role in achieving higher fine-tuning accuracy.

| Layer Range | Avg. | GLUE Benchmark | | | | | | |
|---|---|---|---|---|---|---|---|---|
| | | CoLA | MNLI | MRPC | QNLI | QQP | RTE | SST-2 |
| FT-Full | 81.7 | 59.9 | 84.0 | 85.7 | 90.9 | 90.7 | 68.2 | 92.6 |
| all 12 blocks $[1, 12]$ | 78.0 | 53.6 | 81.4 | 80.6 | 89.3 | **87.2** | 62.8 | **91.0** |
| first 3 blocks $[1, 3]$ | 78.3 | 53.6 | 81.2 | 83.4 | 88.5 | 89.0 | 63.7 | 89.4 |
| last 3 blocks $[10, 12]$ | 76.5 | 48.9 | 79.1 | 81.1 | 87.0 | 86.7 | 61.7 | 90.6 |
| middle 3 blocks $[6, 8]$ | **78.8** | **55.5** | **82.1** | 81.1 | **89.4** | 88.6 | **64.8** | 90.1 |

**Table 6.** Ablation studies on layer range selection with LLaMA-7B (32 blocks) (Touvron et al., 2023) on QA benchmarks. Setting LIFT range to all layers delivers the best accuracy on QA benchmarks.

| Layer Range | QA Benchmarks | | | | |
|---|---|---|---|---|---|
| | PIQA | HellaSwag | SciQ | OpenBookQA | WebQs |
| FT-Full | 82.4 | 59.4 | 95.6 | 47.8 | 44.2 |
| all 32 blocks $[1, 32]$ | **81.1** | **60.4** | **96.3** | **37.6** | **44.1** |
| first 8 blocks $[1, 8]$ | 71.1 | 28.2 | 77.6 | 24.7 | 14.9 |
| middle 8 blocks $[13, 20]$ | 79.4 | 57.9 | 94.2 | 38.1 | 45.2 |
| last 8 blocks $[25, 32]$ | 78.0 | 59.1 | 90.6 | 36.4 | 31.4 |

trained sequentially. For random, candidates are scheduled with a random order using a permutation with replacement (a.k.a uniform sampling).

In Table 7, we compare three different training schedules depicted above. On BERT level experiments, *random* yields the best performance on and *end2front* delivers the worst precision. When model sizes reach $\sim$ 7B and demonstrate zero-shot abilities, the performance of *random* and *front2end* are roughly the same and both of them are better than *end2front*.

With these observations, we do *not* choose complicated schedules. Instead, we simply set the training schedule to be *front2end* throughout all sized experiments in the main paper.

**Table 7.** Ablation studies on training schedules with BERT. LIFT with *random* schedule yields the best performance on GLUE (Wang et al., 2018) benchmark.

| Training Schedule | Avg. | GLUE Benchmark | | | | | | |
|---|---|---|---|---|---|---|---|---|
| | | CoLA | MNLI | MRPC | QNLI | QQP | RTE | SST-2 |
| front2end | 79.3 | 54.5 | 82.4 | 82.1 | 90.3 | 88.6 | **66.1** | 90.8 |
| end2front | 78.8 | 53.5 | 81.8 | 82.4 | 89.7 | 88.8 | 65.2 | 90.1 |
| random | **79.9** | **56.0** | **83.3** | **83.8** | **90.4** | 89.6 | 65.3 | **90.9** |

## D ITERATION POLICY OF LIFT

We study the effectiveness of iteration policy on LIFT. After determining the layer ranges and training schedule, we assign a total of $\lfloor t/N \rfloor$ fine-tuning iterations to each block, where $t$ is the number of iterations and $N$ is the number of learnable blocks in an epoch. Considering 4 blocks with 8 total iterations, the iterations could be either (i) *cyclic*: "1 2 3 4 1 2 3 4" or (ii) *grouped*: "1 1 2 2 3 3 4 4" . We are interested in which type of iterations would yield higher performance.

We compared the performance of different iteration policies, as shown in Table 9. For BERT, the *grouped* policy demonstrated significantly higher performance than the *cyclic* policy, suggesting that LIFT should assign as much contiguous training time as possible to each layer. On LLaMA-7B, the *grouped* policy did not consistently outperform the *cyclic* policy, especially on tasks with non-ideal zero-shot performance (such as OpenBookQA and WebQs). However, given the similar performance, we used the *grouped* iteration policy in our main experiments due to its lower overhead when iterating.

**Table 8.** Ablation studies on training schedules with LLaMA-7B (Touvron et al., 2023) on QA benchmarks. Uniform random yields the best performance when fine-tuning with LIFT on challenging tasks like OpenbookQA and WebQs where zero-shot fails to yield high quality answers. On the other hand, *front2end* and *end2front* perform similarly on the other tasks.

| Training Schedule | QA Benchmarks | | | | |
|---|---|---|---|---|---|
| | PIQA | HellaSwag | SciQ | OpenBookQA | WebQs |
| front2end | **81.1** | 60.4 | **96.3** | 37.6 | 44.1 |
| end2front | 79.3 | **61.5** | 95.9 | 39.8 | 47.9 |
| random | 80.0 | 59.9 | 95.2 | **40.4** | **48.8** |

**Table 9.** Ablation studies on iteration policy with BERT (Devlin et al., 2018). *Grouped* policy consistently outperforms *cyclic* policy on GLUE (Wang et al., 2018) benchmark.

| Iteration Policy | Avg. | GLUE Benchmark | | | | | | |
|---|---|---|---|---|---|---|---|---|
| | | CoLA | MNLI | MRPC | QNLI | QQP | RTE | SST-2 |
| cyclic | 77.4 | 54.4 | 81.8 | 79.9 | 88.6 | 87.4 | 61.2 | 88.3 |
| grouped | **79.9** | **56.0** | **83.3** | **83.8** | **90.4** | **89.6** | **65.3** | **90.9** |

**Table 10.** Ablation studies on iteration policy with LLaMA-7B (Touvron et al., 2023). *Cyclic* and *grouped* policies show their advantages in different tasks.

| Iteration Policy | QA Benchmarks | | | | |
|---|---|---|---|---|---|
| | PIQA | HellaSwag | SciQ | OpenBookQA | WebQs |
| cyclic | 80.8 | **61.8** | 95.7 | **43.4** | **45.5** |
| grouped | **81.1** | 60.4 | **96.3** | 37.6 | 44.1 |

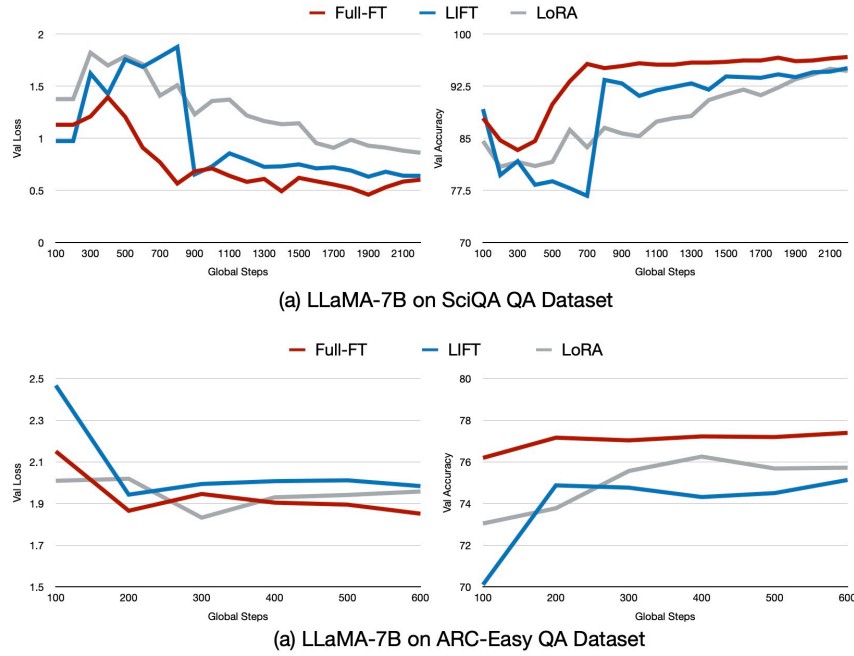

(a) LLaMA-7B on SciQA QA Dataset

(a) LLaMA-7B on ARC-Easy QA Dataset

**Figure 5.** The validation curve of LIFT, LoRA and FT-Full on Llama.