# OpenReview forum: "LIFT: Efficient Layer-wise Fine-tuning for Large Model Models"
_ICLR.cc/2024/Conference — Submitted to ICLR 2024_

### Official Review · Reviewer_6D33 · 2023-10-31

**Soundness:** 2 fair
**Presentation:** 3 good
**Contribution:** 2 fair
**Rating:** 6
**Confidence:** 4

**Summary:**

Current parameter-efficient finetuning methods still suffer from severe efficiency problems even though the number of trainable parameters has been significantly reduced compared to the total number of model parameters. This work owes this problem to the realm that the backward process still needs to be conducted from the last layer to the very front layer, therefore, the computation cost of the backward process remains high with the decrease of trainable parameters. To this end, this work proposes a finetuning method called LIFT to solve the problem. LIFT trains one chosen layer in every training phase, thus, guaranteeing the number of trainable parameters is constrained. This work also tries to mitigate the efficiency problem in the backward process by stopping the backward computation in layers aforehand the chosen layer.

**Strengths:**

The problem this work focuses on is timely and valuable, for the current trend of reducing computation cost during training is to reduce the number of trainable parameters, and this work goes further. The proposed method is simple and easy to apply to existing models and may mitigate the efficiency problem.

**Weaknesses:**

The layer selection strategy seems overly simplified. The role and impact of training each layer in the training process are not thoroughly analyzed; only superficial observations are presented in the article. Additionally, it is questionable whether directly changing one layer of the original model will result in a loss of learning ability since changing the original network may cause greater damage to the original model abilities than other additive methods such as LoRA. Without assessing these issues, it is difficult to justify the actual effectiveness of the approach.

**Questions:**

The opportunity that all parameters in the model can be updated is considered an advantage of LIFT. However, if a subset of layers or parameters is responsible for certain model-learned abilities, such as few-shot learning, changing these parameters during LIFT may bring severe forgetting problems in these abilities. Please help further elaborate on the advantages of iteratively training all model layers and the potential impact of LIFT on the forgetting problem. In addition, the three scheduling schemes explained in the paper may not be able to fully cover the selection of learning sequences, and the impact of different learning sequences of the model layer on learning efficiency and final performance is not clearly analyzed. It would be beneficial if the authors could provide a more specific analysis of this issue.

---

> ### Author Response · Authors · 2023-11-19
>
> Thanks for your time and effort in providing comments on our work. However, we respectfully disagree with the main point that backbones should forever be frozen during fine-tuning. Please refer to our response below.
>
> ### **Q1: Changing the original network may cause greater damage to the original model abilities than other additive methods such as LoRA**
>
> This claim is not correct. Even for LoRA, the low-rank matrices are fused into the original parameters. In practice, people sometimes fuse multiple LoRA weights[0] into one model, thereby changing the original weights. Furthermore, we would like to point out that:
>
> - Besides additive methods, there are also selective methods (BitFit, FishMask), adapter-based methods (Adapters, AdaMix), and reparameterization-based methods (LoRA, KronA) in PEFT, as shown in survey [1]. These methods **do indeed change backbone parameters after fine-tuning**, but they **show comparable performance** in both corresponding benchmarks and zero-shot/few-shot settings.
> - When transferring LLMs to new domains (Finance[2], HealthCare[3]), or equipping LLMs with new abilities (ChatBot[4, 5], LongSequence[6]), **full fine-tuning has been an important baseline** and **has usually demonstrated the best performance**. There is no clear evidence showing that keeping the original network will lead to better generalization.
>
> To address your concerns in more detail, we conducted additional experiments to benchmark the forgetting problems of LIFT and other baseline models. We evaluated the zero-shot abilities of fine-tuned models on other downstream tasks, specifically OpenBookQA and SciQ. The results are presented below:
>
> | PEFT Methods | PIQA | OpenBookQA | SciQ |
> | --- | --- | --- | --- |
> | Original Model | 77.4 | 28.2 | 89.7 |
> | FT-Full  (fine-tuned on PIQA) | 82.4 | 29.2 | 86.9 |
> | LoRA  (fine-tuned on PIQA) | 81.6 | 28.1 | 87.6 |
> | LIFT  (fine-tuned on PIQA) | 81.4 | 28.4 | 87.5 |
>
> As shown in the table, updating all parameters in LIFT does not degrade the zero-shot abilities compared to LoRA.
>
> [0] https://huggingface.co/docs/diffusers/training/lora#working-with-multiple-lora-checkpoints
>
> [1] Scaling Down to Scale Up: A Guide to Parameter-Efficient Fine-Tuning
>
> [2] FinGPT: Open-Source Financial Large Language Models
>
> [3] HuatuoGPT, towards Taming Language Model to Be a Doctor
>
> [4] [https://github.com/tatsu-lab/stanford_alpaca](https://github.com/tatsu-lab/stanford_alpaca)
>
> [5] LIMA: Less Is More for Alignment
>
> [6] Effective Long-Context Scaling of Foundation Models
>
> ### **Q2: strategy seems overly simplified**
>
> We would rather take this as complementary rather than criticizing. Simple yet effective designs make LIFT versatile and applicable to various scenarios. Our results are strong and generally recognized by other reviewers
>
> - #ep71: “thorough evaluation of LIFT across different models”
> - #thJD: “thorough and comprehensive comparison of various tuning methods”
> - #zNBJ: “simple and effective”).
>
> With a “simple” strategy, LIFT demonstrates on-par accuracy with full-finetuning and consistently outperforms previous PEFT methods. This should not be the weakness.
>
> ### **Q3: role and impact of training each layer … not thoroughly analyzed; The three scheduling schemes .. may not be able to fully cover the selection…**
>
> In fact, we have already discussed the **impact of different layers in Figure 4 from Appendix B** and analyzed **different training schedules in Table 7 from Appendix C.**
>
> For layer-wise impact, we found that
>
> - Attention and the first-FFN contribute more to the downstream performance.
> - Middle blocks are more important in evaluating all GLUE tasks.
>
> For training schedules, we would point out that
>
> - There are a total of **N! schemes**, and it is **impossible to exhaustively test them all**. Therefore, we chose the most representative methods (end2front, front2end, and random) for comparison.
> - In BERT level experiments, random yields the best performance, while end2front delivers the worst precision.
> - For llama-7b, the performance of random and front2end is roughly the same, and both outperform end2front.
>
> To ensure LIFT's generalizability and simplicity, we set the training schedule to be *front2end* for all experiments in the main paper. The results show that LIFT with the front2end heuristic achieves comparable accuracy to full fine-tuning.
>
> —
>
> We hope that our responses have addressed your concerns. **Please let us know if there are any other experiments or clarifications we can provide to convince you to increase the rating**.

---

> > ### Author Response · Authors · 2023-11-22
> >
> > Dear reviewer,
> >
> > Thank you once again for providing high-quality reviews. As today is the final day for discussion, could you please let us know if our response has fully answered your questions? We would be more than happy to provide further details and responses for any additional questions you may have.
> >
> > Best regards,
> >
> > Authors

---

### Official Review · Reviewer_tHJd · 2023-11-02

**Soundness:** 3 good
**Presentation:** 3 good
**Contribution:** 2 fair
**Rating:** 5
**Confidence:** 4

**Summary:**

The traditional approach to fine-tuning large language models, while parameter-efficient, often does not lead to significant increases in fine-tuning speed because it still requires a full backward pass during training. To address this inefficiency, a new method called LIFT (Layer-wIse Fine-Tuning) has been proposed. LIFT optimizes one layer at a time, reducing both the number of trainable parameters and the computational cost of backpropagation, leading to improved fine-tuning throughput (2.1x to 2.7x) without sacrificing model quality. When combined with existing methods like LoRA, LIFT enhances both compute- and parameter-efficiency, demonstrating up to 3.7x memory savings and up to 2.7x increased throughput. This method holds particular promise for accelerating the fine-tuning process in larger models and with bigger batch sizes, thus making advanced language models more accessible for diverse applications.

**Strengths:**

The manuscript offers a thorough and comprehensive comparison of various tuning methods, as detailed in Table 1, providing a well-articulated summary of the advantages and disadvantages of established tuning techniques. The impact of the newly proposed LIFT method is examined across different large language models (LLMs), including OPT and LLaMA, demonstrating its broad applicability. Furthermore, Figure 1 effectively highlights the distinct features of LIFT and clearly conveys the rationale behind introducing this new computation-efficient approach.

**Weaknesses:**

The main critique of the proposed LIFT method is the potential for excessive memory usage when applied to multiple downstream tasks, as it necessitates storing separate sets of weights for each task. This could become impractical for practitioners managing numerous tasks, considering the current preference for approaches like LoRA, which require minimal additional storage per task while leveraging a shared, frozen pre-trained model base. LoRA's design also supports batch processing for its minimal weights across tasks.

While the authors have mentioned the possibility of integrating LoRA with their LIFT method, the manuscript lacks detailed exploration of this combination, particularly since the presented experiments focus mainly on LIFT without incorporating LoRA.

To address this concern, the authors are encouraged to provide a more in-depth discussion on handling multiple downstream tasks with the proposed method, possibly extending the authors' experimental framework to include scenarios where LIFT is used in conjunction with LoRA. Such an investigation would not only illustrate the practicality of LIFT in common use cases but also strengthen the argument for its efficiency and versatility when combined with other parameter-efficient methods.

**Questions:**

Please see the Weakness comments

---

> ### Author Response · Authors · 2023-11-19
>
> We sincerely appreciate reviewer #tHJd for acknowledging our evaluation as "thorough and comprehensive" and rating LIFT as a "new computation-efficient approach". We understand the potential concern about parameter-efficiency when serving multiple downstream tasks, and we would like to elaborate on this in more detail.
>
> ### **LIFT is orthogonal with PEFT methods like LoRA**
>
> LIFT is a learning algorithm that can be combined with most PEFTs. Taking the example of LoRA mentioned earlier, we can insert low-rank learnable parameters to blocks (LoRA), and in each iteration, only one LoRA branch is being updated (LIFT). It is worth noting that LIFT does not change the architecture or introduce any new learnable parameters. Therefore, the **additional storage per task for LIFT + LoRA is exactly the same as LoRA-only**. This combination allows us to maintain parameter-efficiency when handling multiple downstream tasks.
>
> ### **The Computation Cost of Fine-tuning Numerous tasks is also considerable.**
>
> While LoRA + LIFT shares the same additional storage per task as LoRA, it is important to note that LoRA does not reduce the cost of fine-tuning numerous tasks. Taking Stanford Alpaca as an example, even with a powerful 8xA100 server, it still takes 5.8 hours to fine-tune LoRA, which is not a significant improvement from FT-Full (7.3 hours). However, with LIFT, it only takes **3.2 hours (2.3x with LIFT)** **or 2.8 hours (2.6x with LIFT+LoRA**). If we have numerous (~1000) customization needs, the savings from LIFT can be quite significant.
>
> While the community has been focusing on parameter efficiency, the efficiency of fine-tuning is rarely discussed. Therefore, we designed LIFT to address this limitation and raise awareness within the community.
>
> ### **Extending Experiments with LoRA + LIFT**
>
> We have evaluated the results of LoRA + LIFT on OPT-1.3B models and Llama-7B, as shown in Table 3 of the submission. To further address your concerns, we have also conducted LoRA + LIFT experiments on BERT (with GLUE) and Llama-7B (with Alpaca).
>
> | BERT-BASE (input 32x512) | Avg score on GLUE | Speed | Learnable Params | Peak Memory |
> | --- | --- | --- | --- | --- |
> | FT-Full | 80.2% | 118.1ms | 125M | 11.3GB |
> | LoRA | 79.0% | 93.1ms | 0.1M | 9.6GB |
> | LIFT | 79.5% | 74.5ms | 8.3M | 9.5GB |
> | LIFT + LoRA | 79.1% | 73.2ms | 0.1M | 9.4GB |
>
> | Llama-7B (input 2x512) | Alpaca Loss | Speed | Peak Learnable Params | Peak Memory |
> | --- | --- | --- | --- | --- |
> | FT-Full | 0.34 | 514ms | 6.7B | 91GB |
> | LoRA | 0.41 | 403ms | 21M | 41GB |
> | LIFT | 0.36 | 221ms | 201M | 32GB |
> | LIFT + LoRA | 0.38 | 194ms | 21M | 27GB |
>
> On both experiments, LIFT + LoRA achieves better performance and speed than LoRA, while using the same number of learnable parameters.
>
> ---
> We hope that our responses have addressed your concerns. **Please let us know if there are any other experiments or clarifications we can provide to convince you to increase the rating**.

---

> > ### Comment · Reviewer_tHJd · 2023-11-22
> > **Response**
> >
> > Thank you for your thoughtful responses and for conducting additional experiments.
> >
> > Although the combination of LoRA and LIFT suggested in your response is intriguing, the main text still maintains that substantial dedicated storage, equivalent to the full model, is necessary. Recent developments in efficient batching methods for various LoRA weights indicate that the proposed method may not support such batching due to its large storage requirements.
> > While I appreciate the value of the additional experimental results, a shift in the manuscript's focus towards PEFT+LIFT would necessitate a resubmission, complete with extensive results and practical applications.
> >
> > Furthermore, the manuscript's use of the term 'latency' is somewhat ambiguous. It's clear that your method may reduce latency during batch processing in the fine-tuning phase. However, this could lead to a misunderstanding that inference latency post-fine-tuning is also diminished. Therefore, it's crucial to clarify the definition of 'latency' to prevent any confusion. It's important to prioritize the quality of fine-tuning, keeping in mind that dedicated storage for each downstream task is vital for the scalability of LLM services.
> >
> > Based on these observations, I maintain my initial evaluation score.

---

> > > ### Author Response · Authors · 2023-11-22
> > >
> > > Thank you for your helpful suggestions. We will  shift our focus to PEFT + LIFT and change the term `latency` to `throughput` for better clarifications.
> > >
> > > LIFT + LoRA is built upon LoRA and does not alter LoRA architectures. Therefore, LIFT + LoRA can also benefit from the development of efficient batching methods for LoRA. Could you kindly provide some references that we can include for comparison in our next version?

---

### Official Review · Reviewer_ep71 · 2023-11-06

**Soundness:** 3 good
**Presentation:** 3 good
**Contribution:** 3 good
**Rating:** 6
**Confidence:** 3

**Summary:**

The paper introduces Layer-wise Fine-Tuning (LIFT), an efficient method for fine-tuning large language models (LLMs). As LLMs grow in size, traditional fine-tuning becomes computationally expensive. While previous studies have focused on parameter efficiency, LIFT aims to improve fine-tuning throughput by learning one layer at a time, thereby reducing both the number of trainable parameters and the depth of back-propagation required. This approach not only saves memory but also increases throughput by 2.1x to 2.7x compared to full fine-tuning, without compromising the final model quality. LIFT is compatible with existing methods like LoRA, and its combination with these can lead to further improvements in compute and parameter efficiency. The paper evaluates LIFT's effectiveness on BERT, GPT, and LLaMA models and discusses its advantages in memory- and compute-constrained environments.

**Strengths:**

1. LIFT presents a novel approach to fine-tuning by updating one layer at a time, which is a creative combination of layer-wise learning and fine-tuning strategies. This method is original in its application to LLMs and addresses the limitations of prior results by reducing computational overhead.
2. The paper provides a thorough evaluation of LIFT across different models, demonstrating its effectiveness in reducing memory usage and increasing throughput. The quality of the research appears to be high, given the detailed comparisons with existing methods and comprehensive ablation studies.
3. The paper is well-structured, with clear explanations of the LIFT method, its implementation, and its advantages over existing fine-tuning methods. The use of figures and tables to compare LIFT with other methods enhances the clarity of the presented information.
4. The significance of LIFT is evident in its potential to make fine-tuning LLMs more accessible by lowering computational barriers. This has broad implications for the field, especially for researchers with limited computational resources.

**Weaknesses:**

1. While LIFT is a novel approach, the paper should discuss how it relates to and differs from the concept of greedy layer-wise unsupervised learning, which also involves layer-by-layer training.
2. The paper would benefit from a more diverse set of experiments, including fine-tuning on a wider range of tasks and datasets to fully understand the generalizability of LIFT.
3. A more detailed comparison with state-of-the-art methods, including those that may not focus on parameter efficiency but achieve high throughput, would be valuable.

**Questions:**

1. How does the selection of layers to fine-tune affect the final model performance? Is there a heuristic for selecting layers that could lead to better results?
2. Combination with Other Methods: The paper mentions that LIFT is orthogonal to methods like LoRA. Could you provide more insights into how LIFT interacts with these methods and any potential limitations of such combinations?
3. Additionally, could the authors elaborate on whether there is potential for the combined use of LIFT with other fine-tuning methods to outperform the application of either approach in isolation?
4. The Figure 1 has no *Right*.
5. Furthermore, the availability of your open-source code is eagerly anticipated.

---

> ### Author Response · Authors · 2023-11-19
>
> We appreciate reviewer #ep71 for acknowledging the novelty of LIFT, rating our experiments as "thorough," and our writing as "well-structured." We are truly thankful for the constructive suggestions, and we have provided our point-to-point response below.
>
> ### **Q1: Difference between LIFT and Greedy Layer-Wise Training**
>
> Hinton[1] and Bengio's[2] work primarily focuses on **unsupervised learning**, **forward and backward only one layer**,  trained with **gradient estimators**, aiming to **find good initialization** for deep belief nets.
>
> While we draw inspiration from this previous work, LIFT is fundamentally different. In LIFT, we focus on **supervised fine-tuning**, **forward the whole model and backward to the selected layer**, trained with **back-propagations**, targeting to **deliver high-performance weights** for large language models.
>
> [1] A fast learning algorithm for deep belief nets.
>
> [2] Greedy Layer-Wise Training of Deep Networks
>
> ### **Q2: Layers selection heuristics**
>
> In fact, we have already discussed the **impact of different layers in Figure 4 from Appendix B** and analyzed **different training schedules in Table 7 from Appendix C.**
>
> For layer-wise impact, we found that
>
> - Attention and the first-FFN contribute more to the downstream performance.
> - Middle blocks are more important in evaluating all GLUE tasks.
>
> For training schedules, we would point out that
>
> - There are a total of **N! schemes**, and it is **impossible to exhaustively test them all**. Therefore, we chose the most representative methods (end2front, front2end, and random) for comparison.
> - In BERT level experiments, random yields the best performance, while end2front delivers the worst precision.
> - For llama-7b, the performance of random and front2end is roughly the same, and both outperform end2front.
>
> To ensure LIFT's generalizability and simplicity, we set the training schedule to be *front2end* for all experiments in the main paper. The results show that LIFT with the front2end heuristic achieves comparable accuracy to full fine-tuning.
>
> ### **Q3: Comparison with fine-tuning efficient PEFT**
>
> We would like to include more experiments but very few of PEFT methods can save backward computations. As shown in survey[1],  only **3 of 26** PEFT methods are capable of saving backward computations and they all introduce extra inference overhead. Therefore, we design LIFT in order to address this limitation and raise awareness within the community.
>
> We include a comparison with LST as suggested by reviewer #zNBj, which adds learnable layers in a highway structure, and compares its performance with llama-7b on PIQA. The results are provided below.
>
> | Methods | PIQA (%) | Inference Latency (ms) | Memory (GB) | Speed (ms/token) |
> | --- | --- | --- | --- | --- |
> | FT-Full | 82.4 | 182.6 | 91 | 284 |
> | LST | 80.7 (-1.7) | 201.5 (+10.3%) | 28 | 102 |
> | LIFT | 81.6 | 182.7 | 32 | 122 |
>
> Though LST shows less peak memory usage, there is a performance drop (**1.7% on PIQA**) and an additional **10.3% inference latency**. Instead, LIFT introduces **no inference overhead** while demonstrating **comparable performance** with FT-Full., showing its advantages when fine-tuning llama.
>
> [1]: Scaling Down to Scale Up: A Guide to Parameter-Efficient Fine-Tuning
>
> ### **Q4: Combination of LIFT with other PEFTs**
>
> LIFT can be combined with most PEFTs. Taking LoRA as an example, we first insert low-rank learnable parameters to blocks (LoRA). In each iteration, only one LoRA branch is updated (LIFT). It is worth noting that LIFT does not alter the architecture or introduce any new learnable parameters. Therefore, the **additional storage per task for LIFT + LoRA is exactly the same as LoRA-only**. This combination allows us to improve fine-tuning throughput while maintaining parameter-efficiency for multiple downstream tasks.
>
> Shown in Table 3, taking the llama experiments with WebQS as an example, LIFT + LoRA achieves higher accuracy than LoRA (39.7 v.s. 33.1) while achieving a much faster speed (106ms v.s 213ms). To further prove the effectiveness, we also evaluate LIFT + LoRA on GLUE and Alpaca dataset:
>
> | BERT-BASE (input 32x512) | Avg score on GLUE | Speed | Learnable Params | Peak Memory |
> | --- | --- | --- | --- | --- |
> | FT-Full | 80.2% | 118.1ms | 125M | 11.3GB |
> | LoRA | 79.0% | 93.1ms | 0.1M | 9.6GB |
> | LIFT | 79.5% | 74.5ms | 8.3M | 9.5GB |
> | LIFT + LoRA | 79.1% | 73.2ms | 0.1M | 9.4GB |
>
> | Llama-7B (input 2x512) | Alpaca Loss | Speed | Peak Learnable Params | Peak Memory |
> | --- | --- | --- | --- | --- |
> | FT-Full | 0.34 | 514ms | 6.7B | 91GB |
> | LoRA | 0.41 | 403ms | 21M | 41GB |
> | LIFT | 0.36 | 221ms | 201M | 32GB |
> | LIFT + LoRA | 0.38 | 194ms | 21M | 27GB |
>
> On both experiments, LIFT + LoRA achieves better performance and speed than LoRA, while using the same number of learnable parameters.
>
> ---
>
> We hope that our response has addressed your concerns. **Please let us know if you have any further questions or comments.**

---

> > ### Author Response · Authors · 2023-11-22
> >
> > Dear reviewer,
> >
> > Thank you once again for providing high-quality reviews. As today is the final day for discussion, could you please let us know if our response has fully answered your questions? We would be more than happy to provide further details and responses for any additional questions you may have.
> >
> > Best regards,
> >
> > Authors

---

### Official Review · Reviewer_zNBj · 2023-11-07

**Soundness:** 2 fair
**Presentation:** 3 good
**Contribution:** 2 fair
**Rating:** 5
**Confidence:** 3

**Summary:**

This paper proposes a memory- and computation-efficient way, LIFT, to fine-tune a large language model. Specifically, LIFT iteratively selects an intermediate layer and then only updates the selected layer during backpropagation. Extensive experiments on natural language processing benchmarks demonstrate the effectiveness of LIFT.

**Strengths:**

- The idea of the proposed LIFT seems to be simple and effective.
- The paper is well-written and easy to follow.

**Weaknesses:**

- As the paper emphasizes memory-efficient fine-tuning, I would expect more comparison to Ladder side tuning (LST)[1]. LST adds a light-weight side module to the large backbone, and only tunes the added side module. Therefore, gradients only need to be back-propagated through the lightweight side module. On the other hand, the proposed LIFT may need to back-propagate the gradients to the front layers in the backbone sometimes. I would expect LIFT’s “peak” memory cost would be higher than LST.
- The paper mentions that, in order for LIFT to save memory, proper implementations might be needed. Does that mean LIFT cannot be efficient on popular frameworks, like PyTorch?

**Questions:**

Please see the weakness section.

---

> ### Author Response · Authors · 2023-11-19
>
> Thanks for your thoughtful comments, and we appreciate constructive advice! We are glad with suggestions to make our experiments more solid and we have conducted experiments for LST. Please check our point-to-point response below.
>
> ### **LIFT Focuses on compute-efficiency rather than memory-efficiency**
>
> The reviewer mentioned, "the paper emphasizes memory-efficient fine-tuning...". This is actually not correct. The main contribution of LIFT is the improvement in fine-tune efficiency. As shown in Figure 2, existing PEFT methods reduce learnable parameters, but this improvement soon saturates for larger workloads, and latency is not significantly reduced. For example, LoRA **saves 100x on trainable parameters**, but is only **1.2x faster**. On the other hand, LIFT improves training efficiency by **2.1x to 2.7x**, making fine-tuning more affordable for broader research communities.
>
> ### **Comparison between LST and LIFT on performance, latency, and memory.**
>
> Thank you for suggesting the comparison with LST. We have mentioned and cited LST in Table 1. Since the original LST only conducted experiments on T5 and not experiment on large language models like GPT and llama, we followed LST's official implementation[1](https://github.com/ylsung/Ladder-Side-Tuning) to conduct experiments on Llama-7B with the PIQA benchmark (bs=2, seq_length=512, measured on A100).
>
> | Methods | PIQA (%) | Inference Latency (ms) | Memory (GB) | Speed (ms/token) |
> | --- | --- | --- | --- | --- |
> | FT-Full | 82.4 | 182.6 | 91 | 284 |
> | LST | 80.7 (-1.7) | 201.5 (+10.3%) | 28 | 102 |
> | LIFT | 81.6 | 182.7 | 32 | 122 |
>
> Though LST shows less peak memory usage, there is a performance drop (**1.7% on PIQA**) and an additional **10.3% inference latency**. Instead, LIFT introduces **no inference overhead** while demonstrating **comparable performance** with FT-Full., showing its advantages when fine-tuning llama.
>
> ### **PyTorch Implementation and Transformers Integration.**
>
> The core part to implement LIFT is to ensure **only the optimizer state of (currently) learnable parameters is stored in GPU memory** (and offload others to CPU). This is a common technique used in algorithms such as  paged optimizer[2], which has been already merged in the Transformer library.
>
> We implemented LIFT **similarly in PyTorch and transformer** and **achieved real savings** (all memory and latency numbers in submission are real-measured). We will open source the codebase and make sure every pytorch user can easily adapt for their need.
>
> [1]: https://github.com/ylsung/Ladder-Side-Tuning
>
> [2]: [https://github.com/huggingface/transformers/pull/23217](https://github.com/huggingface/transformers/pull/23217)
>
> We hope that our responses have addressed your concerns. **Please let us know if there are any other experiments or clarifications we can provide to convince you to increase the rating**.

---

> > ### Author Response · Authors · 2023-11-22
> >
> > Dear reviewer,
> >
> > Thank you once again for providing high-quality reviews. As today is the final day for discussion, could you please let us know if our response has fully answered your questions? We would be more than happy to provide further details and responses for any additional questions you may have.
> >
> > Best regards,
> >
> > Authors

---

### Meta-Review · Area_Chair_4gR4 · 2023-12-08

**Metareview:**

This paper proposes a method for efficient finetuning of large language model. The paper is well written and easy to follow. The proposed method is simple and is verified on different large language models including OPT and LLaMA. However, three of the reviewers are concerned about the compatibility of the proposed method with other methods such as LoRA. Although, the authors provided some additional experimental results by integrating LIFT on LoRA in the response, one reviewer believed that the additional results would lead to a sufficiently new contribution and necessitate a resubmission. Besides, more experiment results, such as the comparison to LST, are expected by reviewers. The ACs agree with the reviewers and believe that more comprehensive evaluations for the method are necessary before publication.

**Justification For Why Not Higher Score:**

More comprehensive evaluations, including the compatibility with LoRA and the comparison with LST, are needed.

**Justification For Why Not Lower Score:**

N/A

---

### Decision · Program_Chairs · 2024-01-16

Reject